# Nuclear Magnetic Resonance and Calorimetric Investigations of Extraction Mode on Flaxseed Gum Composition

**DOI:** 10.3390/polym12112654

**Published:** 2020-11-11

**Authors:** Fang Dubois, Corentin Musa, Benoit Duponchel, Lucette Tidahy, Xavier Sécordel, Isabelle Mallard, François Delattre

**Affiliations:** 1Littoral Côte d’Opale University, UR 4492, UCEIV, Unité de Chimie Environnementale et Interactions sur le Vivant, SFR Condorcet FR CNRS 3417, 145 Avenue Maurice Schumann, 59140 Dunkerque, France; fang.duboisyu@gmail.com (F.D.); corentin.musa@gmail.com (C.M.); lucette.tidahy@univ-littoral.fr (L.T.); isabelle.mallard@univ-littoral.fr (I.M.); 2Littoral Côte d’Opale University, UR 4476, UDSMM, Unité de Dynamique et Structure des Matériaux Moléculaires, 145 Avenue Maurice Schumann, 59140 Dunkerque, France; benoit.duponchel@univ-littoral.fr; 3Littoral Côte d’Opale University, UR 4493, LPCA, IRenE, 145 Avenue Maurice Schumann, 59140 Dunkerque, France; xavier.secordel@univ-littoral.fr

**Keywords:** flaxseed gum, polysaccharide, ultrasonic-assisted extraction

## Abstract

We discussed about the influence of extraction mode on the flaxseed gums composition and their thermal stabilities. In order to do so, flaxseed gum was extracted by both classical magnetic stirring method and ultrasonic-assisted extraction (UAE). As a function of time, protein content, gum yield, pH values were evaluated and samples were characterized by ^1^H and ^13^C nuclear magnetic resonance (NMR) experiments as well as scanning electron microscopy (SEM), differential scanning calorimetry (DSC) and thermal gravimetric analysis (TGA). The flaxseed gum extracted in aqueous solution correspond to a mixture of different components, including polysaccharides, proteins and sometimes lignan derivatives. It is found that the protein and gum contents increase with the extraction duration for both the ultrasonic assisted and the traditional extraction while the pH decreases at the same time. As expected, compared to traditional magnetic stirring method, ultrasonic assisted extraction method can significantly enhance the yield of polysaccharides, lignans and proteins. The variation of pH is correlated to the increase of lignan molecules in the extracted samples. For thermic methods, SEM experiments showed that lignan derivatives which ester-bonded to polysaccharides associated to proteins are responsible to the formation of globular aggregates. Supplementary rod-like molecular organization were obtained from UAE and questions on the nature of the amphiphilic mesogen carbohydrate structures.

## 1. Introduction

With concerns to various environmental issues, materials derived from plants that are biodegradable in nature upon disposal have been paid enormous attention in recent years. Among available raw materials, mucilage or gum is a substance that occurs naturally in some plants, where it acts as the role to keep water. It is a fraction of plant polysaccharides with varying compositions that can become viscous in the presence of water. Therefore, it is also called a translucent and amorphous vegetable hydrocolloid, resulting from the polymerization of a monosaccharide or mixtures of several types of sugars, and many of them combining with uronic acids. So far, the mucilage from many plants has been studied and among them, mucilage from flaxseeds has been shown considerable potential to have wide and promising applications ranging from food industry to cosmetics and medical health care [1,2]. This complex polymeric substance of carbohydrate nature with a highly branched structure contains varying proportions of l-arabinose, d-galactose, l-rhamnose and d-xylose, as well as galacturonic acid. It has been reported that the neutral fraction of flaxseed mucilage contains l-arabinose, d-xylose, and d-galactose, and the acidic fraction contains l-rhamnose, l-fucose, l-galactose and d-galacturonic acid [3,4]. The neutral fraction consists of an arabinoxylan derivative having a (1→4)-*β*-d-xylan backbone to which arabinose and galactose sidechains are attached at positions 2 and/or 3 [3]. The acidic fraction owns a backbone of (1→2)-linked *α*-l-rhamnopyranosyl and (1→4)-linked d-galactopyranosyluronic acid residues, with side-chains of fucose and galactose residues, the former being mostly located at the non-reducing end.

The optimization of flaxseed hydrosoluble compounds extraction (polysaccharides, proteins, mineral salts…) can be monitored by various parameters such as time, temperature, pH as well as stirring velocity [5,6,7,8,9,10]. It has been shown that concentration and composition of gums and consequently physicochemical properties are affected by the operational parameters. Apart from thermic proceeding, ultrasonic assisted extraction (UAE) has been described as the most used eco-friendly technique to improve the extraction efficiency of the mucilage [11]). Three interconnected mechanisms have been described by Mason [12] through which ultrasound can enhance the efficiency of the extraction process by providing better mass transfer: (i) plant cells breaking down via the formation of microjets due to asymmetrical bubble collapsing near a solid surface; (ii) enhanced analytes solubility and solvent penetration due to the increase in localized temperature and pressure in the zone of bubble implosion; and (iii) enhanced diffusion as a consequence of the microstreaming generated by ultrasound. The ultrasonic assisted extraction shows a second order kinetics with respect to mucilage concentration while the extraction by traditional magnetic stirring exhibits first-order kinetics, and therefore an extraction duration of 30 min is reported to be sufficient to achieve quantitative extraction of the mucilage. It is also reported that the recovered mucilage is found to have less proteins and higher concentration of valuable pentoses. Otherwise, Alix et al. [13] have reported that higher amount of protein content was recovered during with the traditional magnetic stirring extraction method at higher temperature. However, no detailed studies were reported on the evolution of the flaxseed gum composition according to different extraction parameters for both traditional and ultrasonic assisted extraction. The involvement of protein extracted with mucilage is not well defined in regard to the performance of mucilage. Furthermore, only few studies have been conducted concerning the thermal properties of flaxseed mucilage. Their thermal stability or their use as eco-friendly thermal plastic films has not yet been investigated.

In this present work, we have executed both the ultrasonic assisted and thermal extraction of flax seed mucilage with varying extraction parameters in order to quantify the protein content extracted together with the mucilage with different extraction parameters as well as its effect on the physical performance of gum. ^1^H and ^13^C nuclear magnetic resonance (NMR) were further conducted to state information about the components presented in the extracted mucilage mixtures. Then, the morphologies of the extracted samples were characterized by scanning electron microscopy (SEM) and the thermal properties of the hydrocolloid mixtures were investigated and discussed by differential scanning calorimetry (DSC), and thermal gravimetric analysis (TGA).

## 2. Materials and Methods

### 2.1. Material and Reagents

Flax seeds (Aramis) produced commercially in North France (Van Robaeys Frères, France) was used as the raw material for the extraction of mucilage. D_2_O was used as NMR solvent for ^1^H and ^13^C NMR characterization. Resorcinol (99%), hydroxydiphenyl solution and l-arabinose were purchased from Sigma-Aldrich (Saint-Louis, MO, United States). Sodium tetraborate (98%), d-galactose (99%) and l-fucose (97%) were purchased from ACROS ORGANICS (Thermo Fisher Scientific Inc., Waltham, MA, USA). d-galacturonic acid and potassium sulfamate (98%) were purchased from Alfa Aesar (Thermo Fisher Scientific GmbH, Kandel, Germany). d-xylose was purchased from VWR chemicals (Fontenay-sous-Bois, France). Sulfuric acid (>95%) was purchased from Fisher chemicals (Fisher Scientific International, Inc., Pittsburgh, Pennsylvania, United States). Bicinchonic acid solution, copper (II) sulfate pentahydrate 4% solution and protein standard (Bovine Serum Albumin-BSA) were used for the protein test and were purchased from Sigma-Aldrich (Saint-Louis, MI, USA).

### 2.2. Mucilage Extraction

Flax seed mucilage was extracted in triplicate by an aqueous process with or without ultrasonic treatment at each desired program. UAE was performed in an ultrasonic pipe equipped with eight radial transducers (13 mm diameter horn, 22 kHz) from SinapTec Ultrasonic Technology (Lezennes, France). The ultrasonic pipe was connected with a NexTgen Ultrasonic Analyzer in order to monitor the developments in real time. The ultrasonic treatment was applied under pulse mode with amplitude of 40% PWM (200W_eff_). All mucilage solutions were collected after each desired extraction duration, followed by centrifugation at a speed of 3500 rpm during 45 min (VWR Mega Star 1.6, VWR International bvba, Leuven, Belgium). Then, the samples were subsequently stored at −80 °C freezer for complete solidification before freeze drying by a freeze-dryer (CHRIST Gamma 2–16 LSCplus, Martin Christ GmbH, Osterode am Harz, Germany). The obtained dry mass was weighted immediately, which is indicated as *W*_dry mass_, and the yield of the total dry mass was calculated by (*W*_dry mass_/70) × 100%.

#### 2.2.1. Traditional Magnetic Stirring Method

For the extraction, 70 g flax seeds were put into 700 mL tap water one hour before the extraction. Then, the extraction temperature was set to room temperature (RT), 40 and 70 °C, respectively with a stirrer speed of 400 rpm. The extraction period was set to 1, 3, 6, 20 and 48 h, respectively.

#### 2.2.2. Ultrasonic Assisted-Extraction

In regard to UAE, a two stages extraction program was conducted. In the first stage, around 70 g flax seeds were put into 700 mL tap water one hour before the extraction process, then the treatment was conducted during one period of 30 and 75 min, respectively. In the second stage, another same amount (700 mL) of tap water was subsequently added into the extraction container after removal of the previous solution, and the successive extraction was conducted during the same period as the previous one.

### 2.3. Flaxseed Gum Characterization

#### 2.3.1. Determination of Sugar and Protein Contents

The neutral sugar composition was determined by colorimetric methods [14,15]. The acidic sugar content was calculated as milligrams of d-galacturonic acid per milligrams of the mucilage [16]. The protein content of each dry mass was tested according to the principle of the bicinchoninic acid (BCA) [17]. Four concentrations (10, 7.5, 5 and 2.5 mg/mL) of each mucilage sample was prepared, and the tests of each sample were all performed in triplicate.

#### 2.3.2. NMR Spectroscopy

All NMR experiments were recorded on a Bruker Avance III spectrometer (Billerica, MA, USA) at 400 MHz (9.4T), equipped with a multinuclear z-gradient BBFO probe head capable of producing magnetic field pulse gradients in the z-direction of 48.15 G.cm^−1^. In standard experiments, the probe temperature was maintained at 298 K and D_2_O was used as solvent.

#### 2.3.3. Scanning Electron Microscopy (SEM)

The surface morphology of mucilage samples was characterized by scanning electron microscopy (SEM) under high vacuum with a field emission SEM instrument (JSM-7100F, Tokyo, Japan). Around 250 mg mucilage samples were first dissolved into around 10 mL distilled water and then the solution was transferred onto a petri dish to allow the evaporation of water, the obtained film was subsequently collected and used for the SEM analysis, and the film sample were coated with a thin (around 10 nm) chromium before observation.

#### 2.3.4. Differential Scanning Calorimetry (DSC)

The differential scanning calorimetry (DSC) thermograms were recorded on a DSC Q1000 DSC instrument (TA instruments, New Castle, PA, USA). An indium standard was used for the calibration and nitrogen was used as the purge gas. The sample was heated from room temperature to 300 °C at a heating rate of 10 °C/min. SDT Q600 (TA Instruments, New Castle, Pennsylvania, United States) was used for thermogravimetric analysis (TGA). Around 4–8 mg samples were scanned from 20 to 800 °C at a heating rate of 10 °C/min in the presence of 100 mL/min nitrogen flow.

#### 2.3.5. Thermal Gravimetric Analysis (TGA)

Thermal decomposition was quantified by the determination of energy activation (*E_a_*) using Broido method [18]. The degradation process of polymer, and a fortiori mixture of polysaccharides, includes various sequences of reaction which depend on the nature of compounds and their intermolecular bonds. Insofar as the pyrolysis give chars and volatiles, the progress of decomposition can be done from thermogravimetric analysis. For an isothermal experiment, the thermal decomposition is given by [19]:(1)$$-\;\frac{d\left(1-\alpha \right)}{dt}=kf{\left(1-\alpha \right)}^{n}$$
where *f* is a function representative of reaction mechanisms, *n* is the reaction order, *k* is the rate constant according to the Arrhenius equation and *α* is the degree of conversion expressed as follow:(2)$$\alpha =\frac{W_{e}}{W_{0}}$$
where *W_e_* is the mass loss by volatilization and *W*_0_ is the initial mass. Thus, the integration and rearrangement of Equation (1) gives Broido equation:(3)$$ln\left[ln\left(\frac{1}{1-\alpha }\right)\right]=\frac{-E_{a}}{RT}+constant$$
where *R* is the universal gas constant, the slope of $ln\left[ln\left(\frac{1}{1-\alpha }\right)\right]$ vs. 1/*T* corresponds to –*E*_a_/*R* and can provide the value of *E*_a_.

## 3. Results and Discussion

### 3.1. Traditional Stirring Extraction

The yield of mucilage, protein content and pH values of the gum solution extracted at 25 °C, 40 °C and 70 °C with varying extraction duration (1, 3, 6, 20 and 48 h) are listed in Table 1. According to the results, both the protein content and yield of mucilage increase along the extraction duration whatever the temperature. As shown in Table 1, the flaxseed gum yield ranges from 3.2% to 6.4% at 25 °C and from 3.9% to 19.9% at 40 °C of seed weight, respectively. Compared to the literature, these values are in a higher range but of the same order to those previously reported [20]. The linearity of extraction yield against temperature between 25 °C and 40 °C highlights the temperature effect on the matter diffusion within the seed. However, the operating mode at 70 °C leads to a drastic raise of yield from the first hour reaching to 8.4%, subsequently exhibits a weaker increase of yields compared to other extraction temperature which is probably due to a partial degradation of flaxseed gum during the extraction.

The yield of protein extracted follows a similar trend to gum extraction, which exhibits an increase as a function of both time and temperature. This phenomenon could be assigned to water molecules, in particular their high probability to be able to cross one and several layers and reach the inside layer “endosperm” and therefore transport water soluble proteins together with the polysaccharides. Furthermore, in order to manage the extraction operation, we have monitored the pH values. These results show an increase of acidity with increasing the extraction duration regardless of the operating temperature.

### 3.2. Ultrasonic Assisted Extraction (UAE)

The mucilage results obtained by UAE with varying extraction duration were summarized in Table 2. According to the results of the first extraction, the yield of total dry mass, yield of mucilage and protein contents increase with increasing the extraction duration, while the pH value decreases slightly and then keeps constant at the same time.

After 30 min of ultrasonic irradiation, the yield of mucilage is close to that obtained from the traditional method at the extraction temperature 70 °C during one hour of extraction. This observation has demonstrated the enhanced efficiency of the ultrasonic assisted extraction compare to the traditional magnetic stirring one. The protein content is in general higher compare to those of the samples obtained by traditional magnetic stirring method for the similar extraction duration. In particular, the protein content of the successive extraction conducted in the second stage is much (around two times) higher than those obtained from the first extraction stage. No significant change of pH was observed during ultrasonic assisted extraction. The results indicate that the increase of protein content in the extracted mixtures was not involved in the decrease of the pH during thermic extraction. This result further supports that other acidic component might be extracted with increasing the extraction duration of the traditional magnetic stirring method.

### 3.3. Determination of Sugar Contents

In order to investigate the evolution of pH value, we further analyzed the composition of the acidic and neutral sugar in the mucilage samples extracted at different temperatures and extraction durations (Table 3). According to the results, we can notice that the acidic and neutral sugar concentration in the mucilage samples decrease with increasing both the extraction duration and extraction temperature. This phenomenon has already been observed [7,9] and was attributed to the degradation of polysaccharide fractions at higher temperature. These observations are consistent with UAE experiments, which are known to enhance polymer degradation [21]. The examination of the neutral and acidic fraction shows lower values compared to those of the literature, indicating that neutral polysaccharide is less sensitive to extraction temperatures. In the case of stirring mode, the ratio of neutral fraction (NF) to acidic fraction (AF) tends to decrease regardless of the application temperature when the extraction duration is more than 20 h, while in the case of UAE, it increases with the successive extraction, which is probably due to the oxidative effects of free radical species on the acidic polysaccharides [22].

The overall pH results indicate that the decrease of the pH value is not caused by the increase of the acidic sugar concentrations, which underlines that other acidic component might be extracted over the extraction procedures.

### 3.4. NMR Characterization

The NMR analysis was subsequently conducted in order to investigate the compounds extracted during the different operating modes. Thus the ^1^H and ^13^C NMR spectra of flaxseed mucilage extracted by traditional magnetic stirring after 1 h and 48 h (extracts obtained after 20 h were presented in Appendix A) and by UAE method were presented in Figure 1 and Figure 2, respectively. According to previously reported literatures [23,24], the ^1^H NMR spectra present three regions ranging from 0 to 3, 3–6 and 6–9 ppm, which can be principally assigned to the protons of the amino acids and carboxylic acids, carbohydrates and phenolic derivatives, respectively (Figure 1a). In a first observation, we notice that the identified parts of spectra can be quite different according to extraction procedure. The analysis of ^1^H NMR data at 25 °C and 40 °C for one hour of extraction show quite similar spectra while with the increase of temperature to 70 °C, the presence of more amino and carboxylic acids as well as phenolic derivatives has been detected. This observation is consistent with the analysis based on pH values that the extracted solutions show decreased pH value with increasing the temperature (Table 1). According to Figure 1a, with increasing the extraction duration from 1 h to 48 h at 25 °C, we can observe the appearance of a singlet at 8.37 ppm suggesting the presence of phenolic acids also called lignans, which are widely distributed in plants and have numerous beneficial effects on human health [25,26]. In general, two classes of phenolic acids can be distinguished: derivatives of benzoic acid and derivatives of cinnamic acid. The hydroxycinnamic acids, consisting mainly of *p*-coumaric, caffeic, ferulic and sinapic acids are more common than the hydroxybenzoic acids. These acids being rarely found in free form, explains the absence of the absorption peaks corresponding to the proton of carboxylic acid in the range of 10–13 ppm [27]. Furthermore, it has also been reported that one of the particularities of the flaxseed is that lignans, such as secoisolariciresinol diglucoside (SDG), are ester-linked via hydroxymethylglutarate (HMG) binding with the carbohydrates to form a macromolecular complex [28,29,30]. Therefore, the absorption peak at 1.25 ppm could be attributed to the absorption of the methyl group of some lignans derivatives extracted from flaxseed. Otherwise, according to the ^13^C NMR spectra shown in Figure 1b, the signature resonance of the ester between 160–190 ppm as well as alkyl group at around 20 ppm are detected for the sample with extraction duration being 48 h, further supporting that the extracted lignan macromolecules are mainly ester-linked with the polysaccharides via HMG bonding. This result is also in accordance with previous analysis based on the pH value, which further supports that the slight decrease of the pH value with increasing the extraction duration for the samples extracted at room temperature is more likely to be caused by the extraction of the lignans while not the acidic sugars of mucilage.

After 48 h of extraction, the resonance of aliphatics and aromatics are mainly predominant for the samples extracted at 40 °C. The aromatic parts reveal peaks from SDG elements as well as from other phenolic compounds, such as *p*-coumaric acid and ferulic acid moieties [31,32]. This is also consistent with the signals corresponding to aromatic carbons in the range of 100–140 ppm and carbonyl group of esters in the range of 160–190 ppm (Figure 1b). Thus, the presence of more phenolic absorption peaks and enhanced peak intensities indicate that more lignan macromolecules can be extracted with long extraction duration at 40 °C compare to room temperature. At 70 °C very weak absorbance peaks on ^1^H NMR spectra in the range of 6–9 ppm could be observed though the intensity of the absorption peak corresponding to the methyl group of HMG at 1.25 ppm increases concomitantly with the extraction duration. On the corresponding ^13^C NMR spectra, the alkyl absorption of HMG is detected for the sample with the extraction duration being 48 h, while no absorption corresponding to the ester bond could be detected. Knowing that the degradation of the lignans in flax seeds was observed at 100 °C [30], it is reasonable to suggest that a lengthy extraction at 70 °C can cause the degradation of lignan molecules and therefore only very weak absorption traces can be detected for the samples extracted at this temperature. This result is also in agreement with the analysis based on the pH value since the presence of the lignans in the sample just after one-hour extraction at 70 °C can be considered to be the reason of a low pH value, and occurrence of degradations of phenolic compounds explains the slight drop of the pH value with increasing the extraction duration at this extraction temperature.

The ^1^H and ^13^C NMR spectra of the samples obtained by two stages UAE with the extraction duration being 30 min and 75 min are shown in Figure 2. According to the results, only just a few traces of absorption spectra corresponding to lignan macromolecules can be observed indicating these compounds can be extracted together with the gum in a short time under sonication though SDG compounds are located in the parenchymatic cells below the mucilage cells [33,34]. This can be assigned to the impact of acoustic cavitation. In regard to the pH value (Table 2), we can explain that the slight increase of acidity of UAE30min-2 compare to that of UAE30min-1 as shown in Table 2 is due to the presence of lignan traces during the second stage of extraction. It is worth to note that the pH value of UAE75min-2 increases slightly compare to that of the UAE75min-1, which indicates a probable degradation of extracted lignans by OH-radicals generated under acoustic cavitation. Indeed, it has been shown that a presence of 5% to 11% of ·OH benzaldehyde derivatives could be produced from hydroxycinnamic acids [35], which might be responsible to the increase of medium basicity. Moreover, another effect of ultrasound irradiation is to extract high concentration of protein which are located in the endosperm of flaxseed.

Otherwise, with referring to the ^13^C NMR spectra, the samples extracted by UAE exhibit very characteristic absorption peaks of the standard reference sugars (see Appendix A) compared to samples obtained by traditional magnetic stirring method. It indicates the possibilities of polysaccharides of mucilage might be degraded into monosaccharide fractions by ultrasound application. Further discussions concerning this point will be conducted in the physical analysis part.

### 3.5. SEM Analysis

In order to obtain general information about the surface morphology of the extracted samples we have conducted SEM analysis on flaxseed (Figure 3). The flaxseed gum is located in the mucous epidermis under the cuticle layer of seed [8]. The examination of unsoaked seed shows the smooth surface of filled epidermis integument pentagonal cells (Figure 3A). At room temperature, the gum extraction and the mass transfer of polysaccharide toward the medium cause the digging of cells viewed by the appearance of the intercellular walls (Figure 3B) as already described elsewhere [36,37]. As we can see on SEM image, the increase of extraction yield induces deeper digging and the edge of the cells is no longer round but more abrupt (Figure 3C). The UAE, which gives a high gum and protein extraction yield, leads to further modification of external seed surface since the cuticle layer seems to have lost of flatness (Figure 3D). These observations highlight the mechanical effect of acoustic cavitation induced by the asymmetric collapse of cavitation bubbles, leading to formation of microjets and thus impacting the flaxseed integument.

Figure 4 shows SEM images of representative textures from extracted samples. Indeed, according to the extraction mode, the thin film of flaxseed gum can exhibit a relative homogenous texture, globular aggregates as well as columnar aggregates as shown in Figure 4A–E respectively. The smooth texture (Figure 4A) was obtained in soft extraction conditions i.e., at room temperature (regardless of the extraction duration) and at 40 °C below 6 h. In these conditions, thin films exhibit nanometric spherical particles (100 to 300 nm) due to the formation of polysaccharide aggregates. The examination of Table 1 allows to correlate the increasing size of aggregates with the decreasing of pH value due to release of hydroxycinnamic acids to give numerous micrometric size spherical particles (Figure 4B). This pH-dependence effect of gelation of flaxseed gum is consistent with previous work since it has been already shown that the gel strength dropped down with decreasing pH values [38]. This is due to synergic coagulation between polysaccharide chains and protein taking place in acidic medium [39,40]. Indeed, it has been demonstrated when pH is higher than pH_i_ (isoelectric point of the protein is pH 4.4) the net charge of the protein becomes negative resulting electrostatic repulsive forces between protein molecules and polysaccharides. At low pH values, the compatibility between polysaccharides and proteins is effective and leads to the formation of aggregates. Thus, with combination to the NMR analysis, it is more reasonable to suggest the macromolecule complexes that formed by the lignan molecules ester-bonded to polysaccharides associated to proteins are responsible to the formation of globular aggregates. However, it is interesting to note that the extracts show a mixed particle size depending on extraction duration at 70 °C since globular particles are more numerous when the extraction is extended, which also corresponds to the increase of phenol diglucoside concentration (Figure 4C). We attribute it to be the positive effect of polyols and sugars against the denaturation of proteins which tends to increase the thermal stability of these macromolecules [41].

SEM images of UAE flaxseed gum (Figure 4D,E) reveals globular aggregates for the whole sonicated samples associated to columnar aggregates whose maximal concentration is reached for longer duration of acoustic irradiation (Figure 4E). This rod-like molecular organization is characteristic of amphitropic liquid crystal which can be obtained from amphiphilic block-copolymer based on glucosides [42]. However, these hierarchical self-assemblies can also come from cyclopeptides who present the adequate ratio between hydrophilic and hydrophobic blocks such as cyclolinopeptides, which are found in flaxseeds [43]. Indeed, it has been shown that sonication is able to increase protein solubility and modify surface hydrophobicity which could favor the formation of supramolecular complexes [44]. These observations imply the nature of the amphiphilic mesogen carbohydrate structures and also the selective extraction to give their specific arrangements. Further investigations will have to be led especially in order to elucidate the potential production of mesogen compounds from flaxseed under sonication. It might be due to that the acoustic cavitation causes the partial degradation of the macromolecule complexes associated to amphiphilic species whose self-assembly forms specific networks.

### 3.6. TGA Analysis

TGA analysis can estimate the thermal stability of a material by measuring the weight loss against the temperature. Figure 5a–e shows TGA and DTG curves of the mucilage sample extracted at room temperature, 40 °C and 70 °C after 1 h and 48 h as well as gum obtained by UAE with extraction duration being 30 and 75 min, respectively. Thermograms exhibit characteristic thermal decompositions of natural gum [45]. The first step of decomposition is related to an initial weight loss due to moisture evaporation (5–11%) between 30–130 °C. It can be noted that the loss of water decreases when increasing the extraction temperature as well as the sonication duration. This indicates a reduction of hydrophilic character of flaxseed gum, which is consistent with the formation of molecular aggregates observed in SEM experiments. The second stage of decomposition appears at 200–350 °C and was due to thermal degradation of both polysaccharides and proteins according to different mechanisms (dehydration, depolymerization or pyrolytic decomposition). After 1 h of extraction at room temperature and 40 °C, the decomposition presents a main peak while in other extraction modes a second process of degradation appears causing either the widening of the transition or appearance of a second band.

For the flaxseed gum samples extracted by UAE, a shoulder peak is detected for all the samples, indicating the higher protein content in the samples extracted by this method. For the samples extracted by traditional magnetic stirring method, the shoulder peak corresponding to proteins is not obvious to be differentiated for the sample extracted at room temperature after 1 h of extraction, probably because the presented amount of proteins (around 7% according to Table 1) is not significant enough for this peak to be appeared in the TGA curve. This result demonstrates that the TGA analysis can also be one of the methods to evaluate the general protein content presented in the extracted mucilage samples.

The *E*_a_ calculated from Broido equation plot (Equation (3)) are shown in Table 4 and Table 5 for thermal methods and UAE, respectively. The thermal decomposition experiments gave low activation energies in the range of 23.9–45.9 KJ/mol.

It can be seen that flaxseed gums extracted at room temperature and 40 °C during one hour have the highest *E*_a_. These results are consistent with thermal stabilities deduced from DSC curves, which will be presented in next section. The hardening of extraction conditions by raise the temperature, lengthening the duration as well as the use of acoustic cavitation cause the decrease of *E*_a_. This finding is in agreement with NMR data which shows a partial degradation of extracted natural compounds and monomer units induced by UAE.

### 3.7. DSC Analysis

Figure 6 shows the thermal heating curves of the samples extracted by both ultrasonic assisted and traditional extraction methods. All thermograms exhibit endothermic transitions whose amplitudes vary depending on molecular mixtures induced by modulation of extraction parameters. According to Figure 5a,b, the main melting peaks of both the samples extracted after 1 and 48 h at room temperature are at around 213–214 °C. The melting peak of the sample extracted after 48 h is narrower and sharper compared to that obtained after 1 h, on which a significant shoulder in the range of 140 and 190 °C can be observed while is almost invisible after a long extraction. Better thermal stability and more homogenous character expand along with the increase of extraction time at room temperature, probably due to the presence of lignan derivatives, which stabilizes the macromolecular mixture to form stable supramolecular aggregates (Figure 6b).

At 40 °C, both the samples show a supplementary lower melting points at around 169 and 157 °C after 1 and 48 h of extraction, respectively, while transitions at around 213–214 °C tend to disappear. These thermograms indicate that samples obtained after longer extraction duration exhibit a decreased thermal stability. At 70 °C, DSC curves present the same trend with a broad transition at around 142 °C and 146 °C after 1 and 48 h, respectively. After one hour of extraction, an unobvious shoulder at around 210 °C can be detected while for a longer time no noticeable shoulder can be detected. The decrease of transition temperatures is consistent to the extraction of numerous macromolecules especially concerning the increase of protein concentrations being likely to evoke transitions associated to the denaturation of these macromolecules [46] and also be probably attributed to the dissociation of the macromolecular entanglement under temperature effect.

The thermal curves of the ultrasonic assisted extractions are shown in Figure 6c. The shape of the DSC heating curves of UAE samples are very similar to the samples obtained by stirring method at 70 °C, which are wider compare to those of the samples obtained at room temperature and 40 °C. This might be due to the degradations caused by both extraction temperature and ultrasound. The mucilage samples are partially degraded into many low molecular fractions leading to the samples with wide and non-uniform compositions and thus lead to broader heating curves. For the 30 min irradiation periods, the endothermic transition is observed at 123 °C for the first stage while it is shifted to 138 °C for the second stage. For a longer irradiation time of 75 min, it is worth to note that the transition of the first stage is maximum at 178 °C then shift to 138 °C during the second stage. This result seems to indicate that a longer duration of acoustic irradiation tends to homogenize the samples whose molecular organizations end up being similar as shown in the SEM images.

The enthalpies of all samples are summarized in Table 6. According to the results, we can note that *E*_a_ of samples are relatively homogeneous for traditional magnetic stirring extraction while UAE experiments show larger deviations in values. The whole values are representative of molecular mixtures extracted form flaxseed and the interpretation of these thermal data are not easy since this depends on molecular self-assembly, dissociation of molecular entanglement following by potential reorganization of supramolecular layout. However, the analysis of thermal UAE data shows a strong disparity of values related to duration and stage. In both cases, the drastic decrease of enthalpy in the second stage can be assigned to the degradation of polysaccharides into monosaccharide fractions.

This DSC analysis demonstrates that the proteins in mucilage samples do not show a clear effect on the thermal behavior of the extracted samples, instead, the extraction parameters do play a key role on the thermal behavior of the extracted samples. Therefore, it reaches the conclusion that the mucilage sample obtained at lower temperature has relatively more uniform compositions and therefore has better physical performance, while those obtained at elevated temperatures or by ultrasonic assisted extraction exhibit relatively broader compositions and therefore exhibit declined physical properties. In addition, based on the DSC analysis, the mucilage sample obtained by traditional magnetic stirring method at room temperature after 48 h of extraction is promising to be used as thermal plastic film, probably because the compositions of the lignan derivatives and the polysaccharides of mucilage seem to be the good ratio to involve ester linked to form macromolecular complexes.

## 4. Conclusions

In the present work, flax seed mucilage was extracted by magnetic stirring and UAE methods. The flaxseed gum extracted in aqueous solution is a mixture of various components, including polysaccharides, proteins and lignan derivatives. Compared to traditional magnetic stirring method, ultrasonic assisted extraction method can significantly enhance the yield of mucilage, proteins and offers the possibility to access at lignan derivatives. It is interesting to find out that the decrease of the pH value of the extracted samples is related to the presence of lignan molecules and could be used as a tracking parameter of SDG derivatives. Effect of the proteins on the physical properties of mucilage is much less compared to that of the lignan molecules, as they can be ester-linked to mucilage polysaccharides via hydroxymethylglutarate (HMG) binding to form macromolecule complexes. Though the extraction yield of mucilage is low, the mucilage sample extracted at room temperature by traditional magnetic stirring method is found to have generally homogenous morphology and exhibit better thermal stability.

Future works are expected to design a scaling up of mucilage UAE in order to develop the great impact of acoustic cavitation on the seed extractions. Indeed, the mixture nature of flaxseed mucilage extraction opens diverse possibilities to develop novel formulas targeted to special applications in cosmetics, food, medical and agriculture field, thus future works are also expected to evaluate the health benefits and pharmacological activities of these mixtures.

## Figures and Tables

**Figure 1 polymers-12-02654-f001:**
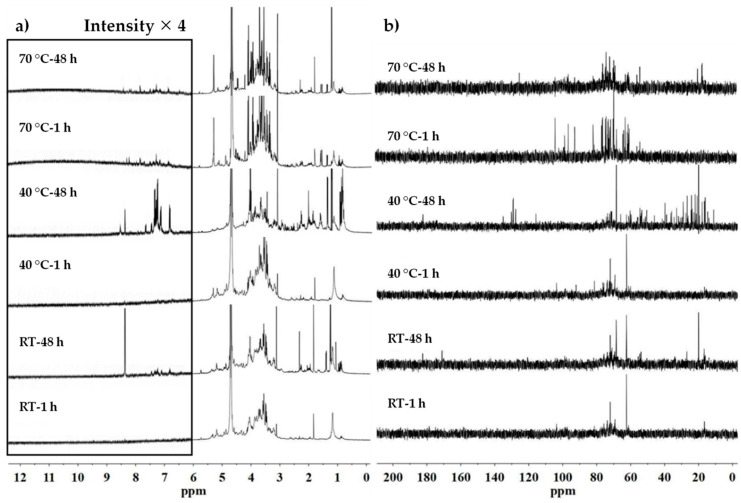
^1^H (**a**) and ^13^C (**b**) NMR spectra of flaxseed mucilage extracted by magnetic stirring method depending on time and temperature.

**Figure 2 polymers-12-02654-f002:**
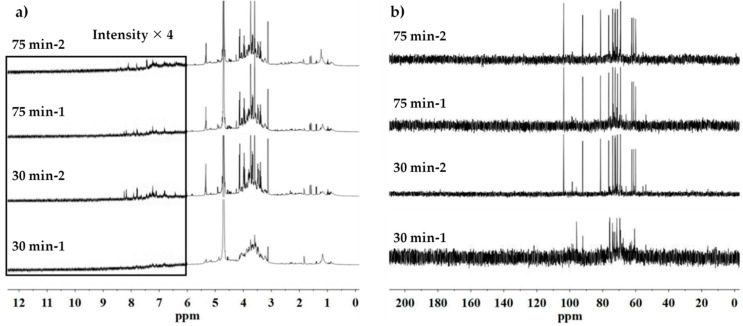
^1^H (**a**) and ^13^C (**b**) NMR spectra of flaxseed mucilage extracted by UAE.

**Figure 3 polymers-12-02654-f003:**
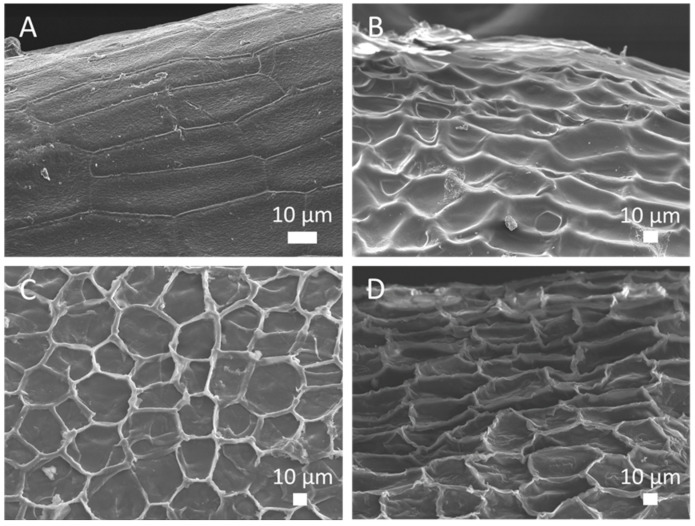
SEM images of surface seeds; unsoaked (**A**) and after extraction during 1 h at 25 °C (**B**), 40 °C (**C**) and UAE75min-1 (**D**).

**Figure 4 polymers-12-02654-f004:**
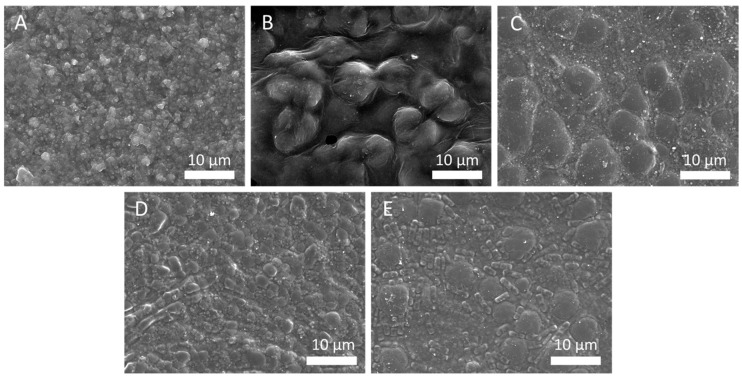
SEM images of mucilage samples extracted at 25 °C over 1 h (**A**), at 40 °C over 48 h (**B**), at 70 °C over 48 h (**C**) and UAE30min-1 (**D**), UAE30min-2 (**E**).

**Figure 5 polymers-12-02654-f005:**
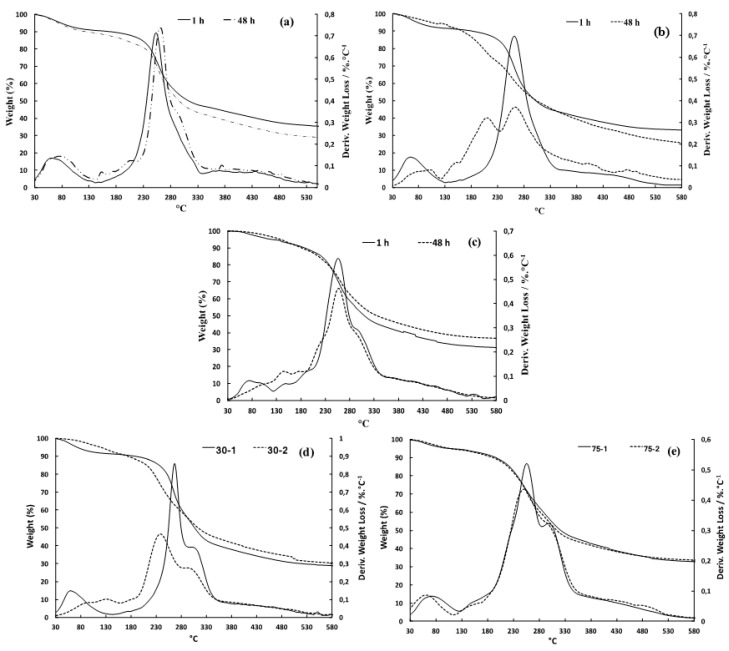
TGA/DTG thermograms of flaxseed gums extracted during 1 h and 48 h at 25 °C (**a**), at 40 °C (**b**), at 70 °C (**c**) and by UAE during 30min (**d**) and 75 min (**e**) for stages 1 and 2.

**Figure 6 polymers-12-02654-f006:**
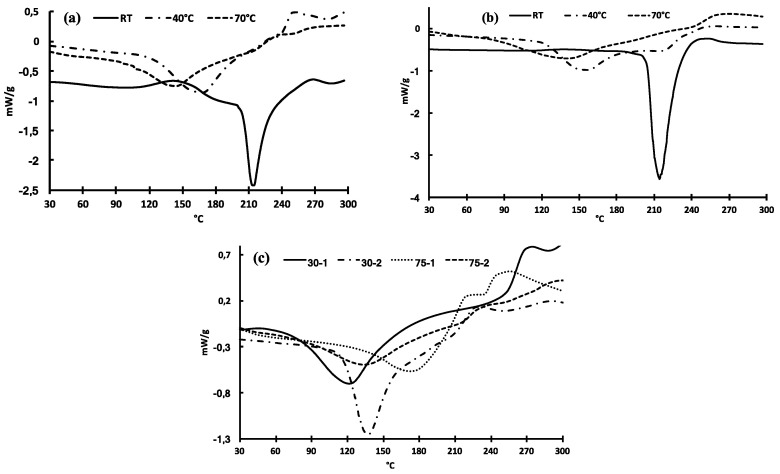
Thermal heating curves of the samples extracted by magnetic stirring mode during 1 h (**a**) and 48 h (**b**) and by UAE (**c**).

**Table 1 polymers-12-02654-t001:** Yield (%), protein content (%) and pH values of flaxseed extracted from traditional stirring extraction.

Extraction Time (h)	Yield of Total Dry Mass (%)			Protein Content (%)			pH		
	25 °C	40 °C	70 °C	25 °C	40 °C	70 °C	25 °C	40 °C	70 °C
**1**	3.2 ± 0.1	3.9 ± 0.2	8.4 ± 0.5	7.0 ± 0.3	9.4 ± 1.2	10.2 ± 0.2	10.1 ± 0.2	9.9 ± 0.1	7.8 ± 0.4
**3**	3.5 ± 0.2	4.7 ± 0.4	10.5 ± 0.7	7.4 ± 1.0	10.4 ± 0.3	15.0 ± 0.8	10.1 ± 0.1	9.5 ± 0.3	7.0 ± 0.1
**6**	4.0 ± 0.3	5.7 ± 0.3	10.9 ± 0.9	9.8 ± 1.6	12.8 ± 0.6	16.2 ± 0.7	8.0 ± 0.1	5.7 ± 0.2	6.9 ± 0.4
**20**	5.8 ± 0.2	8.7 ± 0.6	11.8 ± 0.6	12.3 ± 0.3	16.1 ± 1.9	12.9 ± 0.6	6.7 ± 0.2	5.4 ± 0.2	6.6 ± 0.3
**48**	6.4 ± 0.4	19.9 ± 1.3	12.3 ± 1.1	20.3 ± 2.8	16.1 ± 0.6	17.6 ± 1.2	7.1 ± 0.1	4.8 ± 0.1	6.3 ± 0.2

**Table 2 polymers-12-02654-t002:** Yield (%), protein content (%) and pH values of flaxseed gum from ultrasonic-assisted extraction.

Extraction Period (min)	N of Successive Extraction	Yield of Total Dry Mass (%)	Protein Content (%)	pH
30	1	7.5 ± 0.3	10.5 ± 1.2	8.9
	2	6.7 ± 0.3	20.3 ± 1.4	7.9
75	1	10.9 ± 0.6	20.3 ± 2.2	6.8
	2	4.9 ± 0.1	31.2 ± 1.1	7.8

**Table 3 polymers-12-02654-t003:** Acidic and neutral sugar contents of the selected extraction samples.

Extraction Mode	Extraction Time (h)	Acidic Fraction (AF) (%)	Neutral Fraction (NF) (%)	NF/AF
25 °C	1	16.3	61.2	3.8
	20	15.3	60.6	4.0
	48	14.3	46.7	3.3
40 °C	1	16.1	54.4	3.4
	20	10.5	41.3	3.9
	48	8.4	24.4	2.9
70 °C	1	8.9	42.8	4.8
	20	9.9	56.4	5.7
	48	9.0	45.2	5.0
UAE	0.5	19.3	38.9	2.0
	2 * 0.5	9.4	34.8	3.7
	1.25	15.7	39.6	2.5
	2 * 1.25	8.9	36.4	4.1

* indicates a successive extraction.

**Table 4 polymers-12-02654-t004:** TGA/DTG data of flaxseed gums extracted by magnetic stirring methods.

Extraction Temperature	Extraction Time (h)	N° of Decomposition Stage	Temperature Range (°C)	DTG Maxima (°C)	%wt. Loss	Activation Energy (kJ/mol)
25 °C	1	1	24.7–141.3	61.6	9.3	-
		2	152.4–338.5	253.1	43.5	43.6
	48	1	25.9–142.9	73.0	10.7	-
		2	161.9–357.2	259.3	46.9	38.0
40 °C	1	1	24.7–132.0	64.9	8.4	-
		2	151.4–357.2	261.5	48.0	45.9
	48	1	24.7–126.8	65.2	5.7	-
		2	133.9–358.0	220.2; 260.7	51.7	23.9
70 °C	1	1	27.5–130.4	69.2	5.3	-
		2	146.9–370.3	257.7	52.0	36.1
	48	1	-	-	-	-
		2	190.5–360.4	259.4	41.7	29.0

**Table 5 polymers-12-02654-t005:** TGA/DTG data of flaxseed gums extracted by UAE.

Extraction Time (min)	Extraction Stage	N° of Decomposition Stage	Temperature Range (°C)	DTG Maxima (°C)	%wt. Loss	Activation Energy (kJ/mol)
30	1	1	24.7–143.3	59.2	8.7	-
		2	152.2–379.4	266.3	52.9	40.7
	2	1	28.3–160.3	135.7	8.4	-
		2	163.1–377.4	240.3	48.8	33.0
75	1	1	25.9–132.0	74.1	5.7	-
		2	135.2–381.0	262.2	52.1	34.2
	2	1	24.2–121.1	61.6	5.2	-
		2	135.2–390.3	255.4	52.3	33.8

**Table 6 polymers-12-02654-t006:** DSC characteristics of flaxseed gum extracted by magnetic stirring and UAE methods.

Extraction Temperature	Extraction Time (h)	Temperature Range (°C)	Enthalpy (J/g)	UAE	Temperature Range (°C)	Enthalpy (J/g)
RT	1	141.7–267.7	+318.1	30-1	71.2–268.8	+494.5
	48	139.0–246.3	+379.1	30-2	108.1–226.1	+285.8
40 °C	1	114.5–250.9	+385.2	75-1	88.1–243.4	+318.7
	48	114.5–250.9	+333.4	75-2	88.1–243.4	+170.1
70 °C	1	88.6–236.4	+261.4			
	48	75.2–258.1	+394.7			

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
