# Peer review of "Nuclear Magnetic Resonance and Calorimetric Investigations of Extraction Mode on Flaxseed Gum Composition"

_polymers, 2020, doi:10.3390/polym12112654_

Round 1

Reviewer 1 Report

In this paper, the extraction mode on the flaxseed gum composition was systematically investigated by NMR and calometric. It was found, compared to the traditional method, ultrasonic assisted extraction method could enhance significantly the yield of mucilage, proteins and offered the possibility to access at lignan derivatives. Meanwhile, the presence of lignan molecules was helpful to decrease the pH value of the extracted samples, which could be used as a tracking parameter of SDG derivatives.

The paper was well organized and written, and it was of highly scientific, readable and logical with full and good contents, tables and figures. It could be accepted as an article in the Journal.

Author Response

Thank sir for your comments.

Reviewer 2 Report

The manuscript by Dubois et al entitled: “NMR and calorimetric investigations of extraction  mode on flaxseed gum composition” concerns experiments aimed at characterization of mucilage extracted from flax seeds in diverse conditions. Authors implemented different times, temperatures and involvement of ultrasounds to obtain different mucilage samples which they further analyzed by means of NMR, DSC and TGA. Moreover, they  analyzed samples by scanning electron microscopy. The manuscript is interesting and well written. Please find my minor comments below:

  1. Page 6, lines 223-224 “In the case of stirring mode, the ratio of NF to AF tends to decrease regardless of the application temperature” – abbreviations NF and AF should be explained when used for the first time. Similarly, the abbreviation should be explained in Table 3.
  2. Page 6, lines 225-226: “while in the case of UAE, it increases with the successive extraction, which is probably due to the oxidative effects of free radical species on the acidic polysaccharides” please provide appropriate literature reference.
  3. In Figure 5 letter (b) indicating appropriate thermograms is not present.
  4. Minor English corrections are required: Page 7 line 236: “the 1H NMR spectra presents”, page 2 line 78: “1H and 13C NMR was further”, page 16 line 464: “This DSC analysis demonstrates that the proteins in the mucilage samples doesn’t show”.

Author Response

All suggested corrections have been done and a appropriate litterature added.

Best regards,

Pr. François Delattre
